# Scrutinizing Clinical Biomarkers in a Large Cohort of Patients with Lyme Disease and Other Tick-Borne Infections

**DOI:** 10.3390/microorganisms12020380

**Published:** 2024-02-12

**Authors:** David Xi, Kunal Garg, John S. Lambert, Minha Rajput-Ray, Anne Madigan, Gordana Avramovic, Leona Gilbert

**Affiliations:** 1School of Medicine, University College Dublin, D04 V1W8 Dublin, Ireland; davidxjy.97@gmail.com (D.X.); jlambert@mater.ie (J.S.L.); gavramovic@mater.ie (G.A.); 2Te?ted Oy, 40100 Jyväskylä, Finland; kunal.garg@tezted.com; 3Infectious Diseases Department, Mater Misericordiae University Hospital, D07 R2WY Dublin, Ireland; 4Infectious Diseases Department, The Rotunda Hospital, D01 P5W9 Dublin, Ireland; 5Curaidh Clinic: Innovative Solutions for Pain, Chronic Disease and Work Health, Perth PH2 8EH, UK; drminha@curaidh.com

**Keywords:** Lyme disease, chronic Lyme disease, tick-borne infections, tick-borne co-infections, CD57+ natural killer cells, CD19+ lymphocytes, CD3+ lymphocytes, CD8+ lymphocytes, CD4+ lymphocytes, biomarkers, transferrin

## Abstract

Standard clinical markers can improve tick-borne infection (TBI) diagnoses. We investigated immune and other clinical biomarkers in 110 patients clinically diagnosed with TBIs before (T0) and after antibiotic treatment (T2). At T0, both the initial observation group and patients without seroconversion for tick-borne pathogens exhibited notably low percentages and counts of CD3 percentage (CD3%), CD3+ cells, CD8+ suppressors, CD4 percentage (CD4%), and CD4+ helper cells, with the latter group showing reductions in CD3%, CD3+, and CD8+ counts in approximately 15-22% of cases. Following treatment at the T2 follow-up, patients typically experienced enhancements in their previously low CD3%, CD3+ counts, CD4%, and CD4+ counts; however, there was no notable progress in their low CD8+ counts, and a higher number of patients presented with insufficient transferrin levels. Moreover, among those with negative serology for tick-borne infections, there was an improvement in low CD3% and CD3+ counts, which was more pronounced in patients with deficient transferrin amounts. Among those with CD57+ (*n* = 37) and CD19+ (*n* = 101) lymphocyte analysis, 59.46% of patients had a low CD57+ count, 14.85% had a low CD19 count, and 36.63% had a low CD19 percentage (CD19%). Similar findings were observed concerning low CD57+, CD19+, and CD19% markers for negative TBI serology patients. Overall, this study demonstrates that routine standard clinical markers could assist in a TBI diagnosis.

## 1. Introduction

Misdiagnoses in tick-borne infections (TBIs) are common [1,2,3] and delayed management can result in unnecessary financial costs for the patients [3,4,5,6,7]. Immune and clinical biomarkers can support healthcare professionals in identifying TBIs. Current literature has demonstrated how TBIs can impair the immune system, increasing susceptibility to other infections and prolonging the disease course [8,9,10,11]. The causative pathogens have multiple mechanisms to evade and dysregulate the host immune system [12,13], leading to decreased microbial clearance. Abnormal changes in lymphocyte subpopulations and clinical biomarkers have been associated with Lyme disease and other TBIs [14,15,16,17,18,19]. Laboratory markers can also narrow the differential diagnosis by ruling out conditions with clinical manifestations similar to Lyme disease. Given the increasing incidence of TBIs worldwide [20,21], identifying a series of commonly used clinical markers to aid diagnosis and management can improve patients’ current standard of care.

Current methods to diagnose Lyme disease and other TBIs can be insufficient [11,22,23,24,25,26,27,28,29,30,31]. Lyme disease diagnosis is supported by patient history, clinical examination findings, and completing serology tests [11,23,24,25]. However, patients often cannot remember receiving tick bites or do not develop a bull’s eye rash [11,23,24,25]. Some studies demonstrated that patients with TBIs may have delayed or no seroconversion [22,26,27,28,29,30,31]. Those with immune deficiencies may also have false negative serology results [32]. Routine clinical investigations should be investigated to support current diagnostic algorithms for TBIs. 

TBIs can cause immune dysfunction [9,17,33]. Therefore, abnormal lymphocyte counts and percentages may support TBI diagnoses. Analyses of CD3+, CD4+, and CD8+ lymphocyte subsets can be readily conducted in clinical settings. CD3+, CD8+, and CD4+ T cell lymphocytes are crucial in mounting an effective immune response against infections [34]. Furthermore, CD4+ lymphocytes aid B lymphocytes in affinity maturation and antibody production [34]. Research has shown that mice with B-cell deficiencies had poor outcomes with Lyme arthritis and carditis [35]. CD19+, a B cell-specific antigen essential in B cell signaling and regulation [36,37], could be a potential marker of significance. Previous publications stated that some patients with chronic Lyme disease symptoms had decreased CD57+ natural killer (NK) cell counts [18,19]. Analyzing these immune markers may reveal characteristic biomarker patterns for TBIs. 

Besides immune markers, routine clinical investigations such as total blood count, C-reactive protein (CRP), and transferrin [38] may be abnormal in TBIs. Some of these biomarkers have already been used in disease monitoring [39] and risk stratification [40,41,42]. Additionally, these biomarkers could help exclude conditions frequently confused with TBIs, such as anemia, thyroid, and rheumatological diseases [1,2,3]. Previous investigation into CRP demonstrated it to be significantly elevated in those with post-treatment Lyme disease, compared to those with myalgic encephalomyelitis/chronic fatigue syndrome [43]. Together, these biomarkers can be used to support clinicians in evaluating TBIs.

In this study, we aimed to investigate abnormal immune markers and other clinical biomarkers in a large cohort of patients clinically diagnosed with TBIs at an infectious disease clinic before and after antibiotic treatment.

## 2. Materials and Methods

### 2.1. Patient Cohort

All patients (both sex) over 16 years of age who received consultations at the outpatient infectious diseases clinic at The Mater Misericordiae University Hospital, Ireland, were invited to participate in this study due to suspected tick-borne infections. 

After evaluation by the consultant at the initial visit, T0, 301 patients were clinically diagnosed with TBIs. These patients were found to have non-specific symptoms resembling the flu and a positive history of tick bites or bull’s eye rash, raising the suspicion of TBIs [44]. 

Patients in the initial cohort of 301 were omitted if they only had serology results for *Borrelia burgdorferi sensu lato* species, and patients were only analyzed if they also had conducted serological testing for other co-infections such as *Babesia microti*, *Bartonella henselae*, *Ehrlichia chaffeensis*, *Rickettsia akari*. Additionally, patients were excluded if they had no biomarker investigations conducted at the second follow-up timepoint, T2, and this could be due to missed appointments or symptom resolution that led to the cessation of antibiotic treatment. A final number of 110 patients were analyzed in this study, with results of marker investigation for both T0 and T2, and at T0, patients were prescribed specific antibiotics, as discussed in a previous publication [45]. This longitudinal approach focused on observing changes over time within a consistent cohort, which differed from a retrospective case–control study that should compare cases (individuals with a specific outcome) against controls (individuals without the outcome). As such, there is no clinical case–controls in this study, but reference parameters are outlined in Table 1.

### 2.2. Serology Analysis

Serology analyses for these 110 patients were performed by ArminLabs GmbH (Augsburg, Germany) using TICKPLEX^®^. ArminLabs GmbH is accredited by the German Accreditation Body DAkkS (International Accreditation No. DIN EN ISO 15189:2014) [46]. It is internationally accredited by CAP (College of American Pathologists) and verified by CLIA (Clinical Laboratory Improvement Amendments). TICKPLEX^®^ is a CE IVD (in vitro diagnostic) validated diagnostic test that can detect IgM and IgG antibodies against *Borrelia burgdorferi sensu lato* species, as well as their persistent forms [47]. In addition, TICKPLEX^®^ was used to detect antibodies against other known TBIs, such as *Babesia microti*, *Bartonella henselae*, *Ehrlichia chaffeensis*, *and Rickettsia akari* [47]. 

For analyses, patients were indicated to have either “Positive” or “Negative” antibody responses to the tick-borne infections. Results from the serological testing were compiled into an Excel spreadsheet to handle the data and analysis (Appendix A). Additionally, patients were categorized into these subgroups: *Borrelia* infections only, *Borrelia* and co-infections, Negative Serology, and Non-*Borrelia* Infections.

### 2.3. Analyses of Immune Markers and Biomarkers

Blood samples were collected, and biomarkers were analyzed to identify abnormalities at the initial visit, T0, and the second follow-up, T2. The immune markers examined in this cohort were CD3+, CD8+, CD4+, CD57+, CD19+, cell percentages and counts. We also tested for the ratio of CD4+ Helper T cells to CD8+ Suppressor T cells (H/S ratio), neutrophil count, total lymphocyte count, total white cell count (WCC), total IgG, IgA, and IgM. For each lymphocyte subset, we investigated both lymphocyte percentage and total cell count, as previous studies have shown these results can have different prognostic values in diseases and were not always interchangeable [48,49]. In addition, we conducted laboratory analyses of common clinical biomarkers such as hemoglobin (Hg), platelets, rheumatoid factor (RF), anti-nuclear antibodies (ANA), (CRP), iron, transferrin, transferrin saturation percentage, ferritin, folate, creatine kinase (CK), free thyroxine (FT4), and thyroid stimulating hormone (TSH). 

We calculated the number and proportion of patients with abnormal values in these immune markers or biomarkers within each subgroup. Patient laboratory values higher or lower than the reference values for each subset are considered abnormal. The reference values for each marker can be found in the second row of Appendix A. 

The CD57+ and CD19+ analyses were performed at T0 only. Due to the need to outsource testing for CD57+ and CD19+ lymphocytes with the added cost to patients for those tests, they were made optional at T0. Likewise, not all patients received clinical testing for ANA, RF, FT4, and TSH at T0. The analyses of these markers in this article will reflect the sample size for patients who agreed to have these tests.

Of all immune markers and biomarkers that were analyzed, we focused on key findings of specific markers in this article. The full results are in Appendix A. Appendix A had all Laboratory Results at T0 and T2, respectively. Appendix A provided the amount and percentage of the abnormal biomarkers for the cohort at T1 and T0. Patients were divided into sub-groups of *Borrelia* infections only, *Borrelia* with co-infections, negative serology, and non-*Borrelia* infections, and an analysis of the overall cohort was provided. Appendix A had the results of the CD57+, ANA, CD19+, RF, FT4, and TSH biomarkers at T = 0. 

### 2.4. Statistical Analysis

To compare statistical differences in mean for each marker from T0 to T2 and between patient subgroups, paired *t*-test analyses were conducted using Microsoft Excel 365 and SPSS version 28 [50,51]. The Wilcoxon Signed-Rank test was performed instead for results of each biomarker that were not normally distributed within this cohort [52], as CD8%, IgG, Hg, CPR, and CK were found to not be normally distributed. For sample sizes that were less than 50, the Shapiro–Wilk test was used to test for normality [53]. The Kolmogorov–Smirnov test was used for bigger sample sizes of more than 50 [53]. *p*-values were used to identify the significance of our results, and we defined statistical significance as a *p*-value of less than 0.05, with smaller *p*-values of 0.01 or 0.001 used to indicate more substantial evidence against the null hypothesis [50,51,54]. We also calculated Cohen’s *d*-effect size to help us evaluate and quantify the magnitude of the standardized difference between the means [55]. Effect sizes of d ≥ 0.2, d ≥ 0.5, d ≥ 0.8, and d ≥ 1 were considered small, medium, large, and very large, respectively [55]. A Cohen’s *d* of less than 0.2 is negligible [55]. We included markers that were statistically significant in this article. The complete analysis can be found in Appendix A for the comprehensive analysis and then analysis for the *Borrelia* only subgroup, *Borrelia* and co-infection subgroup, non-*Borrelia* infection subgroup, and for the negative serology subgroup. Fisher’s exact test was also used to determine if the proportional changes in incidences of abnormal markers were significant from T0 to T2 in the cohort of 110 patients [56]. Although 110 patients may seem to be a small cohort, 27 individual parameters for each patient were analyzed with additional 7 parameters for subset of patients: CD57+ (*n* = 37), ANA (*n* = 22), CD19+ (*n* = 101), RF (*n* = 100), FT (*n* = 106) and TSH (*n* = 105).

**Table 1 microorganisms-12-00380-t001:** Marker comparison of 110 patients between T0 and T2.

Marker	Mean		Test Statistic ^3^	Cohen’s *d*	Cohen’s *d* Interpretation ^1^
	T0	T2			
**CD3% (%)** **(reference range: 61–84%)**	**68.00**	69.34	−2.515 **	0.25	Small
CD4% (%) (reference range: 32–60%)	43.76	46.47	−4.742 ***	0.48	Small
CD4+ Helper T Cell Count (reference range: 540–1600)	895.86	943.19	−2.638 **	0.27	Small
H/S Ratio ^2^ (reference range: 0.90–4.50)	2.36	2.47	−2.676 **	0.27	Small
IgG (reference range: 6.00–16.00)	11.19	10.68	1.782 *	0.22	Small
Platelets (× 10^9^ /L) (reference range: 150–400)	269.62	249.27	4.005 ***	0.39	Small
White Cell Count (× 10^9^ /L) (reference range: 3.50–11.00)	6.32	6.03	2.191 *	0.21	Small
Transferrin (g/dL) (reference range: 1.88–3.02)	2.53	2.13	13.113 ***	1.27	Large
Transferrin Saturation (%) (reference range: 19–55%)	28.95	34.48	−4.447 ***	0.44	Small

* *p*-value < 0.05, ** *p*-value < 0.01, *** *p*-value < 0.001, ^1^ Cohen’s d ≥ 0.2, d ≥ 0.5, d ≥ 0.8, and d ≥ 1 were considered small, medium, large, and very large effect sizes, respectively [55]. ^2^ H/S Ratio is the ratio of CD4+ Helper cells to CD8+ Suppressor cells. ^3^ Paired *t*-test is used if the sample data is normally distributed, and Wilcoxon signed rank test is used for samples with non-normal distributions.

### 2.5. Ethics Approval

Ethics approval for the study protocol was granted by the Institutional Review Board of the Mater Misericordiae University Hospital with Institutional Review Board Reference number 1/378/1946. This study complies with the EU CT Directive 2001/20/EC, GCP Commission Directive 2005/28/EC, ICH/GCP, Declaration of Helsinki (1996 Version), and all other applicable local and international regulatory requirements.

## 3. Results

### 3.1. Patient Serology Results

From the serological analysis of all 110 participants, 25 (22.73%) patients had antibodies against *Borrelia* only, while 32 (29.09%) were positive for antibodies to *Borrelia* and other tick-borne co-infections. There were 12 (10.91%) patients that had TBIs that excluded *Borrelia*. Serology testing showed that 41 (37.27%) patients had no antibody responses against TBIs.

### 3.2. Abnormal Markers at T0 and T2 in the Cohort of 110 Patients

Patients at both T0 and T2 had low CD3+, CD4+, and CD8+ lymphocyte cell counts and percentages (Figure 1). Among these results at T0, most patients had low CD3% and low CD8 Suppressor T cell count (20 patients, 18.18%). At T2, low CD8 Suppressor T cell count (20 patients, 18.18%) continued having the highest incidence of patients sharing this abnormality. Using Fisher’s exact test, there was a proportionately significant change (*p*-value = 0.0169) in the incidence of patients with low CD4% from T0 (13 patients, 11.82%) to T2 (three patients, 2.73%). Patients with low (three patients, 2.73%, at both T0 and T2) or high (one patient, 0.91%, two patients, 1.82%) WCC remained proportionately constant at both T0 and T2. Although abnormalities in IgG and IgM levels were observed in this cohort at both T0 and T2, low IgM had the highest incidences at both T0 (13 patients, 11.82%) and T2 (10 patients, 9.09%).

For iron studies at T0, two (1.82%) patients had low transferrin, and 21 (19.09%) patients had low transferrin saturation percentages. In contrast, results from T2 demonstrated greater incidences of deficient transferrin (30 patients, 27.27%) and lower incidences of low transferrin saturation percentage (nine patients, 8.18%). Fisher’s exact tests calculated that the changes in the proportion of patients with deficient transferrin (*p*-value < 0.00001) and low transferrin saturation percentage (*p*-value 0.0294) were statistically significant. The decrease in the incidence of patients with high transferrin from T0 (15 patients, 13.64%) to T2 (three patients, 2.73%) was also proportionately significant (*p*-value = 0.0054)

CRP, a marker of inflammation commonly used in the clinical workup for many diseases [39,40,41,42], did not have the highest incidence of abnormalities in both T0 (14 patients, 12.73%) and T2 (17 patients, 15.45%). 

The full results can be viewed in Appendix A. 

There was a statistically significant increase in the CD3% mean (*t* = −2.515, *p* < 0.01) from 68.00% at T0 to 69.34% at T2 (Table 2). The Cohen’s *d* is 0.25 which indicates a small effect size change. Similarly, the increase in the mean of CD4% from 43.76% at T0 to 46.47% at T2 was statistically significant (*t* = −4.742, *p* < 0.001). Again, this was a small effect size shift with a Cohen’s *d* of 0.48. The mean CD4+ Helper T cell count also statistically increased (*t* = −2.638, *p* < 0.01) from 895.86 at T0 to 943.19 at T2. This change is small as determined by Cohen’s *d* of 0.27. The mean H/S Ratio, which is the ratio of CD4+ Helper T cells to CD8+ Suppressor T cells, also increased significantly (*t* = −2.676, *p* < 0.01) from 2.36 to 2.47 between the two time points. The calculated Cohen’s *d* for this marker is 0.27, which signified a small effect size change in the results. The mean white cell count decreased significantly (*t* = 2.191, *p* < 0.05) from 6.32 to 6.03 between T0 and T2. Cohen’s *d* interpreted this change as small, with a test value of 0.21. The mean total IgG decreased significantly (*t* = 1.782, *p* < 0.05) from 11.19 (T0) to 10.68 (T2). A small effect size change was observed with a Cohen’s *d* of 0.22.

The platelet results demonstrated a decrease in platelet count from 269.62 × 10^9^/L to 249.27 × 10^9^/L when analyzed from T0 to T2. A paired *t*-test calculated a *t*-value of 4.005, which was significant using a *p*-value threshold of <0.001. Cohen’s *d* calculated a small effect size change with a test value 0.39. 

The decrease in mean transferrin from 2.53 to 2.13 between the two time points was statistically significant (*t* = 13.113) using a threshold *p*-value of 0.001. A Cohen’s *d* of 1.27 indicated the magnitude of the effect size was large. Mean transferrin saturation increased from 28.95% (T0) to 34.48% (T2). Using a *p*-value threshold of 0.001, this was considered significant, but Cohen’s *d* calculated the magnitude of this difference was negligible.

The complete statistical analysis can be seen in Appendix A. 

In each subgroup marker abnormalities were observed in all four TBI subgroups, including the negative serology subgroup (Figure 2). Low CD3%, low CD4%, high CRP, low transferrin at T2, and low transferrin saturation at T0 were abnormalities with the highest incidences among all tested markers. The full results are in Appendix A.

In all subgroups, there was an increase in the incidence of patients with low transferrin and a decreased incidence of patients with low transferrin saturation, from T0 to T2. At T0, the incidences of patients with deficient transferrin for each group were: 1 of 25 (4.00%) patients who were classified in the *Borrelia* Infections Only group, 1 of 32 (3.13%) patients from the *Borrelia* and Co-infections group, 0 patients within the TBIs Without *Borrelia* and Negative Serology groups. However, in T2, these proportions grew to 9 out of 25 (36.00%) patients under the *Borrelia* Infections Only group, 7 of 32 (21.88%) patients from the *Borrelia* and Co-infections group, 3 of 12 (25.00%) patients that had TBIs excluding *Borrelia*, and 11 of 41 (26.83%) patients who had no antibodies against the tested TBIs. For the change in incidence of patients with low transferrin saturation, there was a decrease within the *Borrelia* Infections Only group, from 4 (16.00%) to 1 (4.00%). For the *Borrelia* and Co-infections group, there was a decrease from 6 (18.75%) to 4 (12.50%) patients. For patients with non-*Borrelia* TBI antibodies, the incidence decreased from 4 (33.33%) to 2 (16.67%) patients. Lastly, for those in the Negative Serology group, the incidence decreased from 7 (17.07%) patients in T0 to 2 (4.88%) patients in T2. 

All four subgroups had statistically significant decreases in mean transferrin from T0 to T2, using a *p*-value threshold of 0.001 (Table 2). Regarding the *Borrelia* Infections Only group, the mean decreased from 2.58 g/dL to 2.11 g/dL, with a paired *t*-test statistic of 6.186. Using analyzed *Borrelia* and Co-infections group data, mean transferrin decreased from 2.46 g/dL to 2.14 g/dL (*t* = 5.752). Comparing mean transferrin results within the TBIs without the *Borrelia* group, the value decreased from 2.56 g/dL to 2.15 g/dL (*t* = 5.813). Cohen’s *d* calculated large effect sizes in the magnitude of change in all three groups. In the Negative Serology group, mean transferrin decreased from 2.55 g/dL to 2.14 g/dL from T0 to T2.

Mean transferrin saturation increased for all groups except patients from the TBIs Without *Borrelia* group. This increase is more substantial within the *Borrelia* Infections Only group, shifting from 27.66% at T0 to 36.69% at T2 (*t* = −2.971, *p* < 0.01). The effect size of this increase was identified as medium, with a Cohen’s *d* of 0.59.

Increased mean CD4% was observed in all patient groups except the TBIs Without *Borrelia* group. The increase in mean CD4% was most notable in the *Borrelia* and Co-infections patient group. The mean CD4% increased from 44.84% to 47.92% (*t* = −6.571, *p* < 0.001), and a Cohen’s *d* of 1.31 indicated that this change was considered significant.

### 3.3. Abnormal CD57+, CD19+, ANA and RF Values at T0

Among the 37 patients who had received additional testing for CD57+, 22 (59.46%) patients had low CD57+ counts (Table 3), and low CD57+ counts were observed in 11 (61.11%) of 18 patients in the negative serology subgroup. Of the 22 patients tested for ANA, 5 (22.73%) patients were weakly positive, and one (4.55%) had a positive result. A total of 101 patients were tested for CD19+. Low CD19+ results were observed in 15 (14.85%) patients. Of the four subgroups, patients with only *Borrelia* infections had the most significant percentage of low CD19+ counts (five patients, 21.74%). A total of 37 (36.63%) of 101 patients were tested with low CD19%. When comparing between the TBIs subgroups, the subgroup involving tick-borne co-infections without *Borrelia* had the highest percentage of patients with low CD19% (five patients, 41.67%). Both low CD19+ (six patients, 15.38%) and low CD19% (16 patients, 41.03%) were seen in patients from the negative serology subgroup. For the RF marker, 9 (9.00%) out of 100 patients had high RF. The highest proportion were from the negative serology subgroup (4 of 36 patients, 11.11%).

## 4. Discussion

Our results showed abnormal immune markers in this cohort of 110 patients, both at T0 and T2. Low CD3+, CD4+, CD8+, CD19+, and CD57+ white cell percentages and counts were present in this patient cohort (Figure 1 and Figure 2, with Table 3). Abnormalities in neutrophil count, WCC, IgG, and IgM levels were also detected. Among these results at T0, most patients had low CD3% and low CD8 Suppressor T cell count (20 patients, 18.18%). At T2, low CD8 Suppressor T cell count (20 patients, 18.18%) continued having the highest incidence of patients sharing this abnormality. Using Fisher’s exact test, there was a proportionately significant change in the incidence of patients with low CD4% from T0 (13 patients, 11.82%) to T2 (three patients, 2.73%). These findings could suggest that low CD3% and low CD8 Suppressor T cell count might be used to support suspicion of TBIs within patients. However, CD4% could be more indicative of patients’ responses to antibiotic treatment. T-cell lymphopenia can occur in patients with TBIs like acute Lyme disease [14,15]. CD3, CD8, and CD4 T cell lymphocytes are crucial in mounting an effective immune response against infections [34]. Therefore, low levels of T cell lymphocytes may hinder the body’s ability to combat TBIs. The impact of SARS-CoV-2 virus infection on immune dysfunction has been studied extensively recently [57,58,59,60]. Studies have found T cell lymphopenia in early COVID-19 infection [57,58,59,60] and T cell derangements in those with post-acute COVID-19 syndrome. In the SARS-CoV-2 viral infection, the disease course was more severe with longer recovery if patients had low T cell lymphocyte levels, especially in the CD8 and CD4 subpopulations [59,60]. Other studies also suggest T lymphocytes can prevent disease resolution in TBIs [61,62]. T cells have been suspected in the pathophysiology of chronic Lyme arthritis through inappropriate and prolonged inflammation [61,62]. Research into the exact mechanisms and immune interactions involving T lymphocytes would provide a clearer picture of the pathophysiology of TBIs.

Over half of the patients tested for CD57+ had a low CD57+ count (22 patients, 59.46%) (Table 3), indicating that low CD57+ may be relevant in the pathogenesis of TBIs. Among all 101 patients tested for CD19+, low CD19+ (15 patients, 14.85%) and low CD19% (37 patients, 36.63%) were observed in patients. Low CD19+ may indicate poor disease resolution. Past research discovered that *B. burgdorferi* can cause B cell dysfunction [17], and diminished levels of blood plasmablasts were associated with more prolonged symptoms even after treatment [16]. CD57+ NK cells have more outstanding interferon-γ release capabilities and are more potent in eradicating pathological threats from the body than CD57- NK cells [63]. Therefore, CD57+ NK cell count could potentially impact the severity and duration of chronic symptoms. Previous publications have suggested that low CD57+ cell counts were more prevalent in those with chronic symptoms of Lyme disease rather than those with acute Lyme disease [18,19]. However, discrepancies exist in the interpretation of these immune markers. One study found no significant differences in NK subpopulation cell counts between those with chronic symptoms and those without [64]. As the roles of CD57+ NK cells and CD19 B lymphocytes in TBIs are still inconclusive, further research should be conducted to evaluate their clinical value.

More specialized cytokine profiling would help better understand the immune dysregulation observed in patients with TBIs. Research by Soloski and colleagues distinguished two different cytokine signatures in early Lyme disease that were correlated with seroconversion status, results of liver function tests, and clinical presentations [65]. In SARS-CoV-2 infection, which shares certain similarities with Lyme disease, including lymphopenia in the early stages of the disease [57,58] and chronic symptoms [66], cytokine profiling in patients revealed differences that could be associated with symptom severity and a likelihood of developing chronic symptoms [66]. Persistent activation through the CCR5 receptor on monocytes triggering downstream signaling cascades involving platelets and endothelial cells may be responsible for the symptoms of post-acute sequelae of COVID-19 [66,67]. Investigations into immune processes related to TBIs may lead to discovering the mechanisms behind chronic symptoms. Cytokine profiling and clinical biomarkers can differentiate other infections and autoimmune diseases from TBIs. One study demonstrated that cytokine profiles of post-acute sequelae of COVID-19, post-treatment Lyme disease, and myalgic encephalomyelitis-chronic fatigue syndrome were distinct [68]. Future analysis into cytokines could determine how immune interactions affect the disease course of TBIs.

Abnormalities in transferrin and transferrin saturation percentage are notable findings as well. At T0, two (1.82%) patients had low transferrin, and 21 (19.09%) patients had low transferrin saturation percentage. In contrast, results from T2 demonstrated greater incidences of deficient transferrin (30 patients, 27.27%) but lower incidences of low transferrin saturation percentage (nine patients, 8.18%). The decrease in mean transferrin and the increase in mean transferrin saturation percentage from T0 to T2 were statistically significant (*p* < 0.001). A large effect size change in mean transferrin from T0 to T2 was defined using Cohen’s d (Table 2). Transferrin functions as a physiological mechanism that sequesters iron essential to infectious species [69,70]. However, during acute inflammation, transferrin levels are downregulated [71,72]. Studies have suggested iron deficiency could exist even with expected Hg results [73]. In such cases, low transferrin saturation and low ferritin levels could support a diagnosis of iron deficiency, especially if patients suffer from a chronic condition [73,74,75]. Iron is used by bacteria such as *M. tuberculosis* for growth and can even trigger HIV reactivation [76]. In some TBIs like *Borreliosis*, the pathogens have also developed mechanisms to harvest serum iron and iron stores in patients. Increased presentations of low transferrin at T2 compared to T0 could suggest that the patient’s immune function returned to its baseline, and inflammation was induced while combating TBIs. In healthcare settings where resources for specialized testing can be scarce, common clinical investigations, such as iron studies, have the potential to function as surrogate markers.

Our studies also indicate that negative antibody responses to TBIs were too insufficient to rule out TBIs. At both T0 and T2, serologically negative patients had abnormal clinical markers (Figure 2 and Table 3). Other studies have also demonstrated that patients with TBIs may have delayed or no seroconversion [22,26,27]. TBIs can cause immune dysfunction that might prevent an effective B cell response [17]. Additionally, those with iron deficiencies could weaken B cell functions further [77]. Patients might have delayed antibody formation, which could lead to false negative readings on serology analyses. 

Statistically significant changes in the mean for specific markers were observed between T0 and T2 (Table 1 and Table 2). Collectively, they might indicate antibiotic treatment responses in patients with TBIs. For example, we found statistically significant increases in CD3% and CD4% in the *Borrelia* Infections Only group. Additionally, we detected statistically significant decreases in the means of platelet count and neutrophil count. Although the means were still within the reference ranges, our findings suggest the importance of obtaining these common lab investigations to establish baseline values to track disease progress.

Most patients diagnosed with TBIs did not exhibit high CRP values (Figure 1 and Figure 2, with Table 1 and Table 2). CRP, a marker of inflammation commonly used in clinical workup for many diseases [40,41,42], did not have the highest incidence of abnormalities in either T0 or T2 (Figure 1). Additionally, paired *t*-test analyses did not identify significant changes from T0 to T2 within each subgroup. This finding could help to differentiate TBIs from other infections and autoimmune diseases.

## 5. Conclusions

This study demonstrated that abnormal immune markers and biomarkers were present in those clinically diagnosed with TBIs, even with negative serological results. Low CD3+, CD4+, and CD8+ lymphocytes, low IgM antibodies, and low neutrophils were detected at T0 and T2 within this patient cohort. Among those tested for CD57+ cell count, over half had low CD57+ cell counts at T0. Low CD19+ and low CD19% were observed as well at T0. At T0, more patients had a low transferrin percentage and lower transferrin than at T2. These clinical markers could support the diagnosis of TBIs.

## Figures and Tables

**Figure 1 microorganisms-12-00380-f001:**
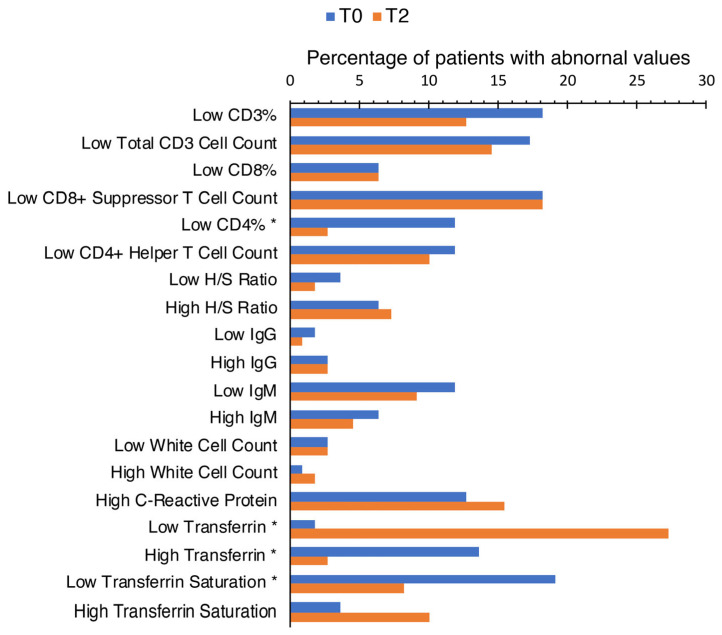
Changes in biomarkers over time points T0 and T2. This figure displays the proportion of patients exhibiting abnormal levels of immunological and iron metabolism biomarkers, specifically low CD4%, low transferrin, high transferrin, and low transferrin saturation at two distinct time points: baseline (T0) and follow-up (T2). Abnormal values are those falling outside of the traditional clinical reference range. A statistically significant difference in these percentages between the baseline and follow-up is noted. Biomarkers with a *p*-value below 0.05, indicating statistical significance as determined by Fisher’s exact test, are marked with an asterisk (*).

**Figure 2 microorganisms-12-00380-f002:**
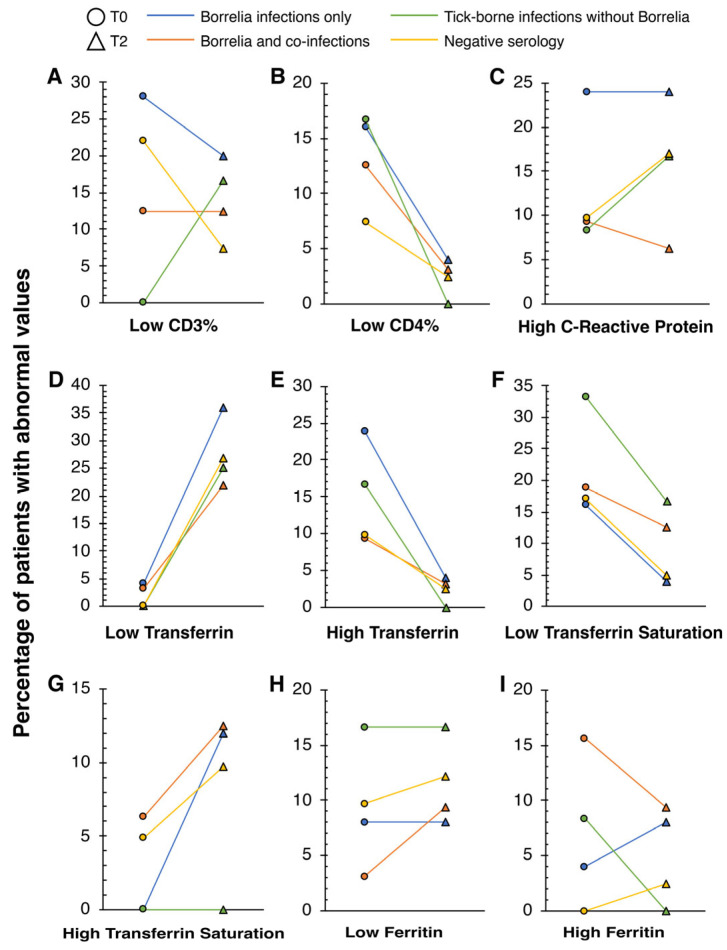
Comparative dynamics of immunological and inflammatory biomarkers in Borrelia-responsive patient groups from baseline (T0) to follow-up (T2). This figure illustrates the variation in the percentage of patients displaying clinically abnormal levels of various immunological and inflammatory biomarkers from T0 to T2. Patient groups are categorized based on their serological response: those with Borrelia-only infections (n = 25), those with Borrelia and other co-infections (n = 32), those with TBIs excluding Borrelia (n = 12), and those with negative serology (n = 41). The biomarkers analyzed include lymphocyte subsets (**A**) low CD3%, (**B**) low CD4%, acute-phase proteins, (**C**) high C-reactive protein, iron metabolism indicators, (**D**) low and (**E**) high transferrin, (**F**) low and (**G**) high transferrin saturation, and storage proteins (**H**) low and (**I**) high ferritin. Abnormal values are defined according to the traditional reference ranges established for each biomarker.

**Table 2 microorganisms-12-00380-t002:** Marker comparison of 110 patients between T0 and T2.

Marker	Mean		Test Statistic ^3^	Cohen’s *d*	Cohen’s *d* Interpretation ^1^
	T0	T2			
CD3% (%) (reference range: 61–84%)	68.00	69.34	−2.515 **	0.25	Small
CD4% (%) (reference range: 32–60%)	43.76	46.47	−4.742 ***	0.48	Small
CD4+ Helper T Cell Count (reference range: 540–1600)	895.86	943.19	−2.638 **	0.27	Small
H/S Ratio ^2^ (reference range: 0.90–4.50)	2.36	2.47	−2.676 **	0.27	Small
IgG (reference range: 6.00–16.00)	11.19	10.68	1.782 *	0.22	Small
Platelets (×10^9^/L) (reference range: 150–400)	269.62	249.27	4.005 ***	0.39	Small
White Cell Count (×10^9^/L) (reference range: 3.50–11.00)	6.32	6.03	2.191 *	0.21	Small
Transferrin (g/dL) (reference range: 1.88–3.02)	2.53	2.13	13.113 ***	1.27	Large
Transferrin Saturation (%) (reference range: 19–55%)	28.95	34.48	−4.447 ***	0.44	Small

* *p*-value < 0.05, ** *p*-value < 0.01, *** *p*-value < 0.001, ^1^ Cohen’s *d* ≥ 0.2, *d* ≥ 0.5, *d* ≥ 0.8, and *d* ≥ 1 were considered small, medium, large, and very large effect sizes, respectively [55], ^2^ H/S Ratio is the ratio of CD4+ Helper cells to CD8+ Suppressor cells, ^3^ Paired *t*-test is used if the sample data is normally distributed, and Wilcoxon Signed-Rank test is used for samples with non-normal distributions.

**Table 3 microorganisms-12-00380-t003:** Total number of patients with abnormal CD57+, CD19+, ANA and RF values at T0.

Total Number of Patients Tested for CD57+ NK Cells, *n* = 37					
	Total	*Borrelia* Infections Only (*n* = 8), N (%)	*Borrelia* and Co-Infections (*n* = 8), N (%)	Tick-Borne Co-Infections without *Borrelia* (*n* = 3), N (%)	Negative Serology (*n* = 18), N (%)
Low CD57+	22 (59.46)	5 (62.50)	5 (62.50)	1 (33.33)	11 (61.11)
High CD57+	0 (0.00)	0 (0.00)	0 (0.00)	0 (0.00)	0 (0.00)
**Total Number of Patients Tested for ANA, *n* = 22**					
	**Total**	** *Borrelia* ** **Infections Only** **(*n* = 7),** **N (%)**	***Borrelia* and** **Co-Infections** **(*n* = 6),** **N (%)**	**Tick-Borne** **Co-Infections** **without *Borrelia*** **(*n* = 4),** **N (%)**	**Negative** **Serology** **(*n* = 5),** **N (%)**
ANA Weak Positive	5 (22.73)	2 (28.57)	2 (33.33)	0 (0.00)	1 (20.00)
ANA Positive	1 (4.55)	0 (0.00)	0 (0.00)	1 (25.00)	0 (0.00)
**Total Number of Patients Tested for CD19 B Lymphocytes, *n* = 101**					
	**Total**	** *Borrelia* ** **Infections Only** **(*n* = 23),** **N (%)**	***Borrelia* and** **Co-Infections** **(*n* = 27),** **N (%)**	**Tick-Borne** **Co-Infections** **without *Borrelia*** **(*n* = 12),** **N (%)**	**Negative** **Serology** **(v = 39),** **N (%)**
Low CD19+	15 (14.85)	5 (21.74)	2 (7.41)	2 (16.67)	6 (15.38)
High CD19+	1 (0.99)	0 (0.00)	1 (3.70)	0 (0.00)	0 (0.00)
Low CD19%	37 (36.63)	9 (39.13)	7 (25.93)	5 (41.67)	16 (41.03)
High CD19%	0 (0.00)	0 (0.00)	0 (0.00)	0 (0.00)	0 (0.00)
**Total Number of Patients Tested for RF, *n* = 100**					
	**Total**	** *Borrelia* ** **Infections Only** **(*n* = 23),** **N (%)**	***Borrelia* and** **Co-Infections** **(*n* = 30),** **N (%)**	**Tick-Borne** **Co-Infections** **without *Borrelia*** **(*n* = 11),** **N (%)**	**Negative** **Serology** **(*n* = 36),** **N (%)**
High RF	9 (9.00)	1 (4.35)	3 (10.00)	1 (9.09)	4 (11.11)

## Data Availability

Tables and Appendix A provide all the relevant data.

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
