# Peer review of "Scrutinizing Clinical Biomarkers in a Large Cohort of Patients with Lyme Disease and Other Tick-Borne Infections"

_microorganisms, 2024, doi:10.3390/microorganisms12020380_

Round 1
Reviewer 1 Report
Comments and Suggestions for Authors
This manuscript has some reference value for the diagnosis of tick-borne diseases, but there are shortcomings:
1. Lack of other clinical case controls caused abnormal CD3+, CD8+, CD4+, CD57+, CD19 and other indicators。
2. There was a lack of biomarker regression analysis or principal component analysis that the authors considered diagnostic。
3.Some table contents can be represented graphically.
4. As a cohort study, the number of cases is relatively small.
Comments on the Quality of English Language/
Author Response
Dear Reviewer 1,
Thank you for your feedback. We have carefully considered your comments and would like to address them as follows:
- Lack of other clinical case controls caused abnormal CD3+, CD8+, CD4+, CD57+, CD19 and other indicators。
Response by authors: Concerning the absence of clinical case controls, this is an inherent aspect of our study's design, as detailed in the methods section. Our longitudinal approach focuses on observing changes over time within a consistent cohort. This differs from retrospective case-control studies that compare cases (individuals with a specific outcome) against controls (individuals without the outcome). Our study, being prospective, monitors temporal variations within a single group, aligning with our defined research objectives. But the reference range of the biomarkers are also provided in Table 2 for more clarity. [Maybe we can clarify this as a limitation in the discussion section]
As such we have put the following in the methods section to make this limitation clearer.
Input of Lines 91-95: This longitudinal approach focused on observing changes over time within a consistent cohort, which differed from a retrospective case-control study that should compare cases (individuals with a specific outcome) against controls (individuals without the outcome). As such, there is no clinical case controls in this study, but reference parameters are outlined in Table 2.
- There was a lack of biomarker regression analysis or principal component analysis that the authors considered diagnostic
Response by authors: Regarding our decision not to employ regression analysis: The R2 value in regression analysis predominantly measures correlation, which is influenced by sample size. Given our study's relatively modest sample size, we conducted an effect size (i.e., sample size independent) analysis utilizing Cohen’s d, as presented in Tables 2 and 4. It is our understanding that this approach provided a more suitable analysis method for our dataset.
- Some table contents can be represented graphically.
Response by authors: We opted for a tabular format to present our data. This decision was based on our belief that the complex and detailed nature of the information would be overly simplified if presented graphically. Tables we believed allowed us to convey the depth and nuance of our findings more effectively.
- As a cohort study, the number of cases is relatively small.
Response by authors: We acknowledge the limited number of cases in our study and have employed statistical tests that take this limitation into consideration. To address this point further, we propose adding a sentence in the method section, explicitly recognizing the small sample size and the power of 26 plus biomarkers analysis performed each of the 110 patients.
Input of lines 179-181: Although 110 patients may seem to be a small cohort, 27 individual parameters for each patient were analyzed with additional 7 parameters for subset of patients: CD57+ (N=37), ANA (N=22), CD19+ (N=101), RF (N=100), FT (N=106) and TSH (N=105).
I hope these clarifications address your concerns effectively. I am open to further discussion and am eager to refine our work based on your insightful feedback.
Reviewer 2 Report
Comments and Suggestions for Authors
The manuscript is well written, very interesting and deserves to be published after minor revision

Author Response
Dear Reviewer 2,
Thank you for your diligent editing and input to the manuscript.
All comments have been address and corrected in the manuscript according to your suggestions. Please see the attached pdf with the acceptance/reply of each of your comments.
Thank you again for your time with this.
